# The Osteogenic Differentiation of Human Dental Pulp Stem Cells through G0/G1 Arrest and the p-ERK/Runx-2 Pathway by Sonic Vibration

**DOI:** 10.3390/ijms221810167

**Published:** 2021-09-21

**Authors:** Won Lee, Su-Rak Eo, Ju-Hye Choi, Yu-Mi Kim, Myeong-Hyun Nam, Young-Kwon Seo

**Affiliations:** 1Department of Plastic and Reconstructive Surgery, Dongguk University Medical Center, Goyang-si 10326, Korea; e1clinic@daum.net (W.L.); surakeo@yahoo.com (S.-R.E.); 2Department of Medical Biotechnology, Dongguk University, Goyang-si 10326, Korea; jooh031919@gmail.com (J.-H.C.); kjmtik@nate.com (Y.-M.K.); iis05047@naver.com (M.-H.N.)

**Keywords:** vibration frequency, dental pulp stem cell, osteogenesis, ERK pathway

## Abstract

Mechanical/physical stimulations modulate tissue metabolism, and this process involves multiple cellular mechanisms, including the secretion of growth factors and the activation of mechano-physically sensitive kinases. Cells and tissue can be modulated through specific vibration-induced changes in cell activity, which depend on the vibration frequency and occur via differential gene expression. However, there are few reports about the effects of medium-magnitude (1.12 g) sonic vibration on the osteogenic differentiation of human dental pulp stem cells (HDPSCs). In this study, we investigated whether medium-magnitude (1.12 g) sonic vibration with a frequency of 30, 45, or 100 Hz could affect the osteogenic differentiation of HDPSCs. Their cell morphology changed to a cuboidal shape at 45 Hz and 100 Hz, but the cells in the other groups were elongated. FACS analysis showed decreased CD 73, CD 90, and CD 105 expression at 45 Hz and 100 Hz. Additionally, the proportions of cells in the G0/G1 phase in the control, 30 Hz, 45 Hz, and 100 Hz groups after vibration were 60.7%, 65.9%, 68.3%, and 66.7%, respectively. The mRNA levels of osteogenic-specific markers, including osteonectin, osteocalcin, BMP-2, ALP, and Runx-2, increased at 45 and 100 Hz, and the ALP and calcium content was elevated in the vibration groups compared with those in the control. Additionally, the western blotting results showed that p-ERK, BSP, osteoprotegerin, and osteonectin proteins were upregulated at 45 Hz compared with the other groups. The vibration groups showed higher ALP and calcium content than the control. Vibration, especially at 100 Hz, increased the number of calcified nodes relative to the control group, as evidenced by von Kossa staining. Immunohistochemical staining demonstrated that type I and III collagen, osteonectin, and osteopontin were upregulated at 45 Hz and 100 Hz. These results suggest that medium magnitude vibration at 45 Hz induces the G0/G1 arrest of HDPSCs through the p-ERK/Runx-2 pathway and can serve as a potent stimulator of differentiation and extracellular matrix production.

## 1. Introduction

Many studies have used media with growth factors, cytokines, and additives to explore the various cell differentiation pathways [1,2,3,4]. A variety of methods and technologies have been applied to increase the activation and differentiation of cells. In particular, mechanical/physical stimulation such as perfusion, stretching, vibration, and compression have been shown to affect the functions of many cell types [5,6,7]. Additionally, many researchers have studied the effects of significant variables such as acceleration, flow rate, frequency, periods, strain, and load [7]. These mechanical/physical stimuli modulate tissue metabolism, and this process involves multiple cellular mechanisms, including the secretion of growth factors and the activation of mechano-physically sensitive kinases.

Clinical studies have shown that vibration of 30 Hz at a low magnitude (0.2 g) inhibits the decline in bone mineral density [8]. Low-intensity ultrasound stimulation (30 mW/cm^2^) applied to tibial defects is highly effective in achieving bone maturation and reducing the time of osteogenesis [9]. Furthermore, Alikhani et al. have reported that mechanical vibration (120 Hz, 0.3 g) inhibits jawbone reduction after tooth extraction. These effects are accompanied by upregulation of osteogenic markers and increased intramembranous bone formation, and downregulation of osteoclastic markers and inflammatory markers alongside decreased bone resorption activity [10]. Additionally, by using an ovariectomy rat model, the same authors have revealed that mechanical vibration (120 Hz, 0.3 g) increases osteoblast activity and decreases osteoclast activity to alleviate osteoporosis [11].

Several in vitro studies searching for the pathways or mechanisms underlying these effects have been conducted using various cells. Prè et al. have reported that mechanical stimulation (30 Hz, 0.6 g) induces high expression of ALP, type I collagen, RUNX-2, osteonectin, and osteocalcin in human mesenchymal stem cells (hMSCs) [12]. Demiray et al. have shown that mechanical vibrations (90 Hz, 0.15 g, 15 min/day) increase Runx2 expression, actin fiber thickness, and the roughness of the cytoplasmic membrane in mouse MSCs undergoing osteogenic differentiation [13]. However, these stimuli do not always promote cellular physiology and differentiation. Baskan et al. have observed that low-intensity vibration (90 Hz, 0.1 g, 15 min/day) downregulates adipogenic markers in mouse MSCs undergoing adipogenic differentiation [14]. Vibration stress of 100 Hz on bone cells has been shown to be significantly worse than that of 5 Hz and 60 Hz in vitro, and this stress is mediated through increased COX-2 expression and NO release [15].

Healthcare products using a variety of mechanical/physical stimuli have been commercialized, such as sonic vibrating toothbrushes, and studies of their effects are being conducted. One investigator reported that various frequencies and intensities of sonic vibrations stimulate the neurological system, bone, spine, and intervertebral discs [5]. Chen et al. showed that 800 Hz sonic vibration (0.30 g) increased the osteogenic differentiation of human MSCs more than 30 and 400 Hz, as assessed via alizarin red staining and mRNA expression analysis [16]. Additionally, osteocytes subjected to sonic vibration (0.3 g) upregulate COX-2 expression 3-fold at 90 Hz but downregulate RANKLE expression by half at 60 Hz [17]. Tirkkonen et al. have reported that 100 Hz sonic vibration (3.0 g) induces the osteogenic differentiation of human adipocyte stem cells by more than 50 Hz, as assessed via alizarin red staining and collagen expression analysis [18]. These results suggest that osteogenesis and bone resorption can be modulated through specific vibration-induced changes in cell activity, which depend on the vibration frequency and occur via differential gene expression. However, there have been few reports of HDPSCs subjected to sonic vibrations.

Many studies have addressed the effects of vibration stimulation on the differentiation of various types of stem cells. Some researchers defined low-magnitude as <1 gravity (g = acceleration of 9.81 m/s^2^) and 3–5 g as high-magnitude [5], but other researchers described a range of 0.3 to 6 g as low-magnitude [6]. Thus, the definitions of low- and high-magnitude have not been standardized. There are also few reports about the effects of 1.12 g magnitude sonic vibrations on the osteogenic differentiation of human dental pulp stem cells (HDPSCs). In our study, we defined 1 to 2.9 g as medium-magnitude.

In this study, we investigated whether medium-magnitude (1.12 g) sonic vibration with a frequency of 30, 45, or 100 Hz could affect the osteogenic differentiation of HDPSCs. The effects of these frequencies on the proliferation and differentiation of HDPSCs were evaluated using FACS, RT-PCR, western blotting, ALP, Ca assays, and immunohistochemical staining.

## 2. Results

### 2.1. Effects of Mechanical Vibration Frequency on HDPSC Proliferation and Stress Level

To test the effect of mechanical vibration on the growth of HDPSCs, cell counting was performed using a Scepter automated cell counter. HDPSCs were cultured in osteogenic differentiation medium, followed by exposure to mechanical vibration for 14 d.

The growth of the cells was significantly inhibited in the mechanical vibration group (Figure 1A). The cell number was determined to be approximately 3.1 ± 0.32 × 10^5^ in the control group, 1.9 ± 0.79 × 10^5^ in the 30 Hz group, 2.1 ± 0.26 × 10^5^ in the 45 Hz group, and 2.0 ± 0.11 × 10^5^ in the 100 Hz group after 14 d of culture. These results indicated that the growth of the mechanical vibration exposed cells decreased as the cells entered differentiation stages faster than the control cells. However, all three mechanical vibration groups exhibited increased LDH secretion compared with the level in the control group (Figure 1B). The results of the LDH assay indicated that 100 Hz of mechanical vibration induced more stress than 30 or 45 Hz.

### 2.2. Effect of Mechanical Vibration Frequency on HDPSC Morphology

Figure 2 shows the morphology of the HDPSCs during osteogenesis at different frequencies on d 7 and d 14. Most of the cells were short spindle and narrow fibroblast-like (Figure 2A–D), but some of the cells were cuboidal (black arrow) (Figure 2C,D) on day 7. On d 14 of culture, many of the cell morphology had changed to a cuboidal shape (black arrow) at 45 Hz and 100 Hz compared to the control groups. These morphological changes indicate that vibration of 45 Hz and 100 Hz induced more osteogenic differentiation compared with the 30 Hz and control groups (Figure 2).

### 2.3. Effect of Mechanical Vibration Frequency on HDPC Surface Antigen Expression

To determine whether or not mechanical vibration altered the HDPSCs surface antigen expression, FACS analysis was performed for CD73, CD90, and CD105, which are known as MSC surface markers. The expression levels of CD73 in the control, 30 Hz, 45 Hz, and 100 Hz groups were 97.8%, 95.9%, 83.1%, and 90.2%, respectively; the CD90 levels were 92.6%, 81.3%, 73.4%, and 77.6%, respectively; and the CD105 levels were 82.4%, 82.3%, 79.4%, and 63.2%, respectively.

The CD73, CD90, and CD105 expression decreased in the mechanical vibration groups compared with the control group. Particularly, the expression of surface antigens was significantly decreased at mechanical vibration of 45 and 100 Hz (Figure 3). This suggests that the appropriate level of mechanical vibration may induce differentiation through changes in HDPSC surface antigen expression.

### 2.4. Effect of Mechanical Vibration Frequency on the HDPC Cell Cycle

The cell cycle distribution was determined by PI staining of the collected cells followed by flow cytometry analysis. As shown in Figure 3, vibration induced an increased proportion of cells in the G0/G1 phase and an accompanying decrease in cells in the S phase. The proportion of the cells at the G0/G1 phase was 60.7% in the control group, 65.9% in the 30 Hz group, 68.3% in the 45 Hz group, and 66.7% in the 100 Hz group after 7 d, whereas 18.7% of the control group were in the S phase, 14.4% of the 30 Hz and 10.4% of the 45 Hz vibration treated cells were in the S phase, and 11.1% in the 100 Hz treated group on day 7 (Figure 4). These results indicate that the inhibition of proliferation of HDPSCs is due to vibration-induced cell cycle G0/G1 arrest (Figure 3).

### 2.5. Effect of Mechanical Vibration Frequency on the Expression of Osteogenic Markers

Figure 4A shows the mRNA levels of osteogenesis-related genes in HDPSCs undergoing osteogenic differentiation with mechanical vibration for 14 d. Osteonectin is a glycoprotein that binds to sodium in the bone and is secreted by osteoblasts during bone formation, initiating mineralization and promoting mineral crystal formation. The osteonectin mRNA levels of the mechanical vibration groups were increased 1.5-fold compared with the level in the control group. In particular, the cells subjected to 45 Hz of mechanical vibration strongly expressed it (2.5-fold increase).

Osteocalcin, BMP-2, and ALP mRNA levels in the 45 and 100 Hz groups were increased 1.5-fold compared with the levels in the control group (Figure 4A). When the in vitro osteogenic differentiation potential of rat dental pulp stem cells was examined, the level of alkaline phosphatase (ALP) activity increased early during induction. Most of the osteoblast markers (osteonectin, osteocalcin, BMP-2, and ALP) were highly expressed in the 45 and 100 Hz mechanical vibration groups. The representative osteogenic marker genes were highly expressed in HDPSCs subjected to 45 Hz, 100 Hz, or 30 Hz vibration. The 30 Hz mechanical vibration group had significantly increased expression of early and late osteoblast marker genes. Additionally, Runx-2 is upregulated by 2.5-fold in the cells subjected to 45 Hz mechanical vibration compared with the levels in the other groups. Thus, this result shows that 45 Hz mechanical vibration affected osteogenic differentiation through Runx-2 (Figure 4A).

As shown in Figure 4B, osteogenesis related proteins (bone sialoprotein, osteoprotegerin, and osteonectin) were increased in the 45 Hz group compared with the other groups. In particular, it had strong BSP protein expression by more than 2.0-fold relative to the control group. Additionally, the osteoprotegerin expression level in the 45 and 100 Hz groups was increased 1.8-fold compared with that in the control group. Additionally, to address potential p-ERK activation during osteogenesis, western blot analysis was performed. In this study, after 14 d of osteogenesis with mechanical vibration, activation of phosphorylated ERK was increased compared with the control group.

### 2.6. Effects of Mechanical Vibration Frequency on ALP and Calcium Deposition

Changes in the ALP level and activity are involved in a variety of physiological and pathological events, such as bone development. A high level of alkaline phosphatase activity indicates good osteogenic differentiation. Jaiswal et al. have reported that ALP activity and calcium deposition of MSCs were increased during osteogenic differentiation [19].

As shown in Figure 5A, the mechanical vibration group showed a higher level of ALP than the control. The ALP levels were similarly elevated at all three mechanical vibration frequencies. These results are consistent with the view that mechanical vibration stimulates osteogenic differentiation.

We investigated the effect of mechanical vibration on the deposition of calcium in HDPSCs during osteogenesis. To confirm the calcium deposition, the media was collected and analyzed by a calcium assay kit. The degree of calcium content was increased in the mechanical vibration groups (30, 45, 100 Hz) compared with the control group (Figure 5B). The black staining of the matrix by von Kossa staining indicates the apparent formation of calcification nodes in response to mechanical vibration, especially at 100 (Figure 5E).

### 2.7. Immunohistochemical Examination

To confirm the capacity of HDPSCs for osteogenic differentiation by mechanical vibration, immunohistochemical analysis was performed. Osteopontin and osteonectin are expressed by osteoblasts during bone formation and are involved in mineralization. In the mechanical vibration groups, osteopontin and osteonectin were more avidly stained than in the control group. In addition, 100 Hz of mechanical vibration was most effective in inducing the expression of osteogenesis proteins. Collagen type I was strongly detected at 45 Hz and 100 Hz, and collagen type III was stained strongly at 45 Hz. Additionally, osteopontin was strongly expressed at 45 Hz and 100 Hz, and osteonectin staining was most pronounced at 100 Hz (Figure 6).

## 3. Discussion

Mechanical stimulation is one of the important enhancing factors influencing osteoblast metabolism. In this study, the frequency effect (30, 45, and 100 Hz) on osteogenesis was evaluated at a medium-magnitude (1.12 g).

When evaluating cell growth, mechanical vibration applied at 30, 45, and 100 Hz reduced HDPSC proliferation. This is similar to the findings of another study in which cell proliferation was decreased by 30 Hz of mechanical vibration [12]. Another study tested the effects of micro-vibrations on the proliferation and osteodifferentiation of bone marrow-derived mesenchymal stromal cells seeded on human bone-derived scaffolds. They showed that cell proliferation was decreased on 7 and 10 d after vibration treatment at 40 Hz [20]. Furthermore, they also found that after low-magnitude, high-frequency mechanical vibration at 10–180 Hz, the proliferation of periodontal ligament stem cells was decreased except at 10 Hz [21].

These stimulations promote cell differentiation but can also cause stress. LDH is a cytoplasmic catalytic enzyme related to the reversible conversion between pyruvic acid and lactic acid and it is released through the cell membrane when the cell is damaged or under stress. Thus, less LDH release means less cellular damage or stress. In this study, HDPSCs were cultured in osteogenic differentiation medium and subjected to mechanical vibration for 14 d. The mechanical vibration groups had an increase in LDH release in proportion to the increase in frequency (Figure 1B). The results of the LDH assay indicated that 100 Hz of mechanical vibration induced more stress than 30 and 45 Hz. Similar results have been reported by other researchers. Bacabac et al. have reported that the vibration stress of 100 Hz was significantly larger than the stress of 5 Hz and 60 Hz and it increased COX-2 expression and NO release [15], and Lau et al. showed that vibration (0.3 g) increased COX-2 expression the most, at 90 Hz [17]. Therefore, high frequency can cause stress, so it is a stimulus that needs to be considered carefully before application.

Usually, HDPSC morphology is fibroblast-like, but some of the cells changed to a cuboidal or polygonal shape in the mechanical vibration groups at 14 d. It was previously reported that DPSCs showed a fibroblastic-elongated morphology at 7 d in differentiation media [22], and other studies have reported altered polygonal morphology of DPSCs after 21 d in osteogenic differentiation media [23]. In addition, Demiray et al. have shown that a polygonal morphological change can be induced by mechanical vibrations (90 Hz, 0.15 g) of mouse MSCs during osteogenesis [13]. However, in this study, such a morphological change was not as dramatic, perhaps due to the short cell differentiation time.

After the HDPCSs were stimulated by mechanical vibration, the expression of all three surface markers was down-regulated at 45 and 100 Hz. Previously, we demonstrated that the expression of CD90 and CD105 on bone marrow MSCs was appreciably down-regulated by mechanical stimulation at 15% tension compared to 5–10% tension [24]. Based on this knowledge, we presumed that mechanical stimulation would synergistically reduce the expression of some surface antigens during osteogenic differentiation. In our previous study, isolated cells expressed MSC surface antigens including CD73 (99.88%), CD90 (99.81%), and CD105 (99.87%) (data not shown). The disappearance of CD73, CD90, and CD105 antigen expression during osteogenesis suggests that these proteins may be involved in the regulation of osteogenesis. So, their down-regulated expression may be a common feature of HDPSC differentiated mesodermal lineages.

In addition, we analyzed the effect of vibration stimulation on the G0/G1 phase of the cell cycle in HDPSCs. The results of this study indicated that the vibration induced an increased proportion of cells in G0/G1 compared with the control group, and a decrease in the proportion of cells in S and G2/M phases.

In a related study, some investigators reported that cell cycle parameters are closely related to cell specification and differentiation. Liu et al. showed that exposure to sevoflurane caused inhibition of the Wnt/β-catenin pathway, cell cycle arrest in the G0/G1 phase, and an earlier switch from proliferation to differentiation. They showed by FACS that the proportion of cells in the G0/G1 phase in the experimental (sevoflurane addition) group showed a statistically significant increase compared with the control group, whereas the proportion of cells in S phase in the experimental group was significantly lower than in the control group. Therefore, their research suggested that sevoflurane arrested the cell cycle at the G0/G1 phase through inhibition of the Wnt/β-catenin signaling pathway, thus resulting in premature differentiation of neural stem cells [25].

In our study, based on the western blot, fluorescence, and CD marker results, we can conclude that differentiation was more strongly induced at 45 and 100 Hz. The proportion of cells in G0/G1 was the highest at 45 Hz. Additionally, the control group was observed to be in the proliferative phase and the proportion of cells in G2/M was the highest in the control at 17.5% (Figure 3). This suggests that sonic vibration affects the cell cycle and reduces proliferation by arresting the cell cycle, an effect more pronounced at 45 Hz compared to the other groups. In the PI staining results, sub-G1, which is generally thought to indicate apoptosis, was increased at 45 Hz and 100 Hz. However, some researchers have reported that sub-G1 may also indicate the cells are differentiating [26]. We did not observe apoptosis or necrosis of the cells under a microscope.

Given the established facts that cell-cycle progression and differentiation are two distinct and mutually exclusive processes [27] and that increases in cell differentiation are generally accompanied by a parallel reduction in cell growth [28], the arrest of cell growth by vibration implies that it can accelerate the osteogenic differentiation of HDPSCs.

In addition, analysis of mRNA and protein expression were performed to evaluate the osteogenic effects of sonic vibration. Based on the mRNA and western blotting results, the peak levels of osteonectin, osteocalcin, BMP-2, ALP, Runx-2, bone sialoprotein, osteoprotegerin, and p-ERK expression occurred in the 45 Hz and 100 Hz groups. Osteocalcin is secreted solely by osteoblasts, is thought to play a role in the body’s metabolic regulation and is pro-osteoblastic, or bone-building, by nature [29]. BMP-2, another bone morphogenetic protein, plays an important role in the development of bone and cartilage. BMP-2 induces osteoblastic differentiation by acting directly on mesenchymal stem cells (MSCs) and is used in the clinic to induce bone formation, albeit at high doses [30]. Additionally, Runx-2, which is involved in the production of bone matrix proteins, is able to up-regulate the expression of major bone matrix protein genes leading to an increase in immature osteoblasts forming from stem cells [31,32]. Bone sialoprotein is a highly post-translationally modified acidic phospho-protein normally expressed in mineralized tissues, such as bone and dentin [33]. Many studies have reported that p-ERK activation is an essential mediator of proliferation and differentiation in various cell types, including osteoblasts [34,35,36,37].

Our gene expression results for HDPSCs during osteogenic differentiation are similar to previous results reported by Zhao et al. concerning the mRNA expression levels of osteocalcin, osteonectin, and type I collagen. Uzer et al. reported that RUNX-2 expression is similar at 30 and 100 Hz at a medium-magnitude (1 g) [38]. Another investigator showed that 30 Hz of vibration (0.6 g) induced mRNA expression of ALP, type I collagen, RUNX-2, osteonectin, and osteocalcin compared with the control group [12]. Alikhani et al. reported that mechanical vibration (120 Hz, 0.3 g) increased mRNA expression of ALP, BMP-2, osteonectin, Osterix, Runx-2, and Wnt-3a. They revealed that mechanical vibration increased bone density by Runx-2 activity mediated through the Wnt-3a/catenin pathway [11].

However, our study showed that 45 and 100 Hz of sonic vibration (1.12 g) increased osteogenic differentiation caused by Runx-2 activity through the BMP-2 and p-ERK pathway. Some researchers reported that low-magnitude mechanical vibration (35 Hz, 0.3 g, 15 min/day) upregulated BMP-2, osteocalcin, Runx-2, and p-ERK in an in vivo study with an ovariectomized rat model. Additionally, another investigator showed that 40 Hz mechanical vibration (0.3 g, 30 min/12 h) upregulated ALP, osteocalcin, Runx-2, and p-ERK in MSCs cultured in a 3D scaffold [39].

In addition to the mRNA and protein levels of osteogenic markers, we observed that the calcium and ALP contents were increased in response to sonic vibration (Figure 5). However, the differences in these contents among the different frequencies were not measured. Nevertheless, the von Kossa staining results showed more calcium deposition at 100 Hz (Figure 5F). In a related study, Prè et al. have shown that mechanical stimulation (30 Hz, 0.6 g) induces high expression of ALP, as assessed by alizarin red staining [12]. Additionally, Uzer et al. have reported that alizarin red staining is more intense in cells subjected to 100 Hz vibration than in those subjected to 30 Hz vibration at a medium-magnitude (1 g) [38]. Another study compared the ALP activity and mineralization between 50 and 100 Hz of sonic vibration (3.0 g). The ALP activities of both vibration groups were increased relative to the control group, and alizarin red staining was the most intense at 100 Hz [18].

Here, we used von Kossa staining because it allowed for the identification of mineral deposition after mechanical vibration (Figure 5C–F). Cells cultured in osteogenic medium without any vibration (control group) exhibited a small number of black particles. The HDPSCs treated with vibrations exhibited matrix mineralization compared to the controls on day 14. In most studies, black mineral deposits were stained only after more than 21 d of differentiation [40]. However, in this study, the cell inoculation concentration was low and the differentiation period was 14 d, so the staining was weak, but even so, similar to the calcium assay results (Figure 5B), the vibration groups were more heavily stained compared to the non-vibration group (control), especially in the 100 Hz group (Figure 5E).

The ALP activity may not vary much among the different frequencies, but the von Kossa staining confirmed there was more calcium deposition at a higher frequency. However, high frequency is not always effective for osteogenesis. Some researchers reported that 90 Hz of vibration increased COX-2 expression and 60 Hz of vibration (0.3 g) reduced RANKLE expression. Thus, 60 Hz vibration can inhibit osteoclastogenesis during bone regeneration [17].

Additionally, our immunohistochemical analysis data demonstrated prominent upregulation of type I collagen, osteopontin, and osteonectin at 45 and 100 Hz. It is well known that osteonectin initiates mineralization and promotes mineral crystal formation and that osteopontin binds to the mineral matrix of bones (Figure 6). Our immunostaining results did not show any obvious difference in protein expression between 45 Hz and 100 Hz, but we observed that it was stained more strongly at 45 Hz and 100 Hz than at 30 Hz (Table 1).

This result implies that the efficacy of mechanical vibration for inducing differentiation is strongly dependent on the frequency of the applied mechanical vibration.

## 4. Materials and Methods

### 4.1. Culture of HDPSCs

HDPSCs were purchased from Lonza (PT-5025, swiss, Basel, Switzerland) and maintained in DPSC BulletKit (PT-3927, PT-4516, Lonza, swiss, Basel, Switzerland) in an incubator at 37 °C with a humidified atmosphere containing 5% CO_2_. HDPSCs were used from passages 4–5, both of which gave similar results.

### 4.2. Stimulation by Mechanical Vibration

The mechanical vibrator used is depicted in Figure 7. An apparatus purchased from Sonic World (Gangneung, Korea) was used to produce mechanical vibration. The controller consisted of the following two major parts: a common signal generator and amplifier, and a regulator of frequency and voltage. The frequency range of the signal generator was set between 1 Hz and 500 Hz and could be freely controlled according to the experimental protocol (Figure 7A). The four channels of the vibrator were made using the principles of speakers. The magnetic field formed on a round coil with an electrical current moved a magnet located in the coil up and down. By adjusting the parameters of the electricity current, the equipment could precisely control the frequency, intensity, and acceleration/deceleration of the vibration induced by the moving magnet. The four channels of the vibrator were connected to an incubator at 37 °C, and a 6-well culture dish with the cells was placed on a black plate in this incubator and covered with an acrylic cover (Figure 7B).

To measure the effect of mechanical vibration on cell differentiation, cells were adhered to the plate containing the osteogenic differentiation medium and then stimulated on the first day by mechanical vibration (11 m/s^2^, 1.12 g, 30 min/day) at a frequency of 30 Hz, 45 Hz, or 100 Hz. The osteogenic differentiation medium consisted of α-MEM with 10% FBS, 10 mM β-glycerophosphate, 50 μM L-ascorbate 2-phosphate, and 10^−7^ M dexamethasone (all from Sigma-Aldrich, St Louis, MO, USA). The medium was changed every 2–3 d for 14 d. To measure the effect of the mechanical vibration, the cells were detached following the stimulation and then analyzed.

### 4.3. Proliferation and Lactate Dehydrogenase (LDH) Assays

Cell proliferation was measured using an automated cell counter (Scepter™, Millipore Corporation, Billerica, MA, USA). A sensor probe was attached to the end of the Scepter unit with the electrode sensing panel facing toward the front of the device.

The cultured cells were washed with PBS and detached with accutase for 10 min. After centrifugation at 1000× *g*, the supernatant was discarded and the cell pellet was resuspended in 1 mL PBS. About 50 µL of the sample was taken up by the probe of the Scepter. The device displayed a histogram of the cell size or diameter on its screen, in addition to as the cell concentration per mL.

LDH activity was measured using a Lactate Assay Kit (Abcam, Cambridge, MA, USA) according to the instructions of the manufacturer. In the 1st step, the frozen reagent was thawed. The lactate probe solution was thawed in a 37 °C water bath for 1–5 min and the lactate enzyme mix dissolved in 220 µL lactate assay buffer. In the 2nd step, we prepared a reaction solution. This reaction solution was composed of 46 µL of lactate assay buffer, 2 µL of probe, and 2 µL of enzyme. After 14 d of culture, 50 µL of media and 50 µL of the reaction solution were mixed and incubated in the dark at room temperature for 30 min. The reaction was terminated by adding the stop solution (1 N HCl), and the absorbance was measured at 570 nm.

### 4.4. Cell Surface Antigen Expression Analysis via Fluorescence-Activated Cell Sorting (FACS)

Antibodies against human antigen CD73 and CD90 were purchased from BD Sciences (San Jose, CA, USA). Anti-CD105 antibody was purchased from Ancell (Bayport, MN, USA). A total of 5 × 10^5^ cells were re-suspended in 200 μL PBS and incubated with a fluorescein isothiocyanate (FITC)- or phycoerythrin (PE)-conjugated antibody for 20 min at room temperature (or for 45 min at 4 °C). The fluorescence intensity of the cells was evaluated via flow cytometry using a FACScan apparatus (BD Sciences, San Jose, CA, USA), and the data were analyzed using CELLQUEST software (BD Sciences, San Jose, CA, USA).

### 4.5. Cell Cycle Analysis

The proportion of cells in each cell-cycle phase was determined via flow cytometry. The HDPSCs were treated with vibration for 7 d and then fixed in 70% ethanol at 4 °C for 20 min. Subsequently, they were incubated in 5 μL of propidium iodide (1 mg/mL, Thermo Scientific, P3566, Waltham, MA, USA) solution, and then in 2 μL RNase A (10 mg/mL, Thermo Scientific, EN0531) at room temperature for 5 min. The percentage of cells in each stage was measured using ModFIT software (BD Biosciences, San Jose, CA, USA). At least 10,000 cells were analyzed for each datum point.

### 4.6. Reverse Transcription-Polymerase Chain Reaction (RT-PCR)

Total RNA was isolated using 1 mL of TRIzol (Invitrogen, 15596026, Waltham, MA, USA). Subsequently, 100 μL of chloroform (Merck, 288306, Darmstadt, Germany) was added, and the solution was mixed and then incubated for 3 min. After centrifugation at 14,000× *g* and 4 °C for 20 min, the upper phase was transferred to a new tube, and 300 μL of isopropanol was added. After an incubation period of 10 min and another centrifugation step of 15 min at 14,000× *g*, the supernatant was discarded. The pellet was washed with 1 mL of 70% ethanol and centrifuged at 4 °C for 5 min. The supernatant was discarded, and the pellet was air-dried. After adding 50 μL of diethylpyrocarbonate-treated water, the pellet was dissolved on ice for 10 min. Total RNA concentration was measured using a Nano Drop device (BioTek, Cytation 3 imaging reader, Winooski, VT, USA). 

Reverse transcription (RT) was performed using 1 μg of the total RNA. The RNA samples were first mixed with 0.5 μg of oligo(dT)20 (Invitrogen, Carlsbad, CA, USA), and 10 mM dNTPs (Invitrogen, Carlsbad, CA, USA). The mixture was then heated at 64 °C for 5 min in a thermocycler to disrupt any secondary RNA structures and afterward cooled to 4 °C for 1 min. Subsequently, 4 μL of first-strand buffer, 1 μL of 0.1 M DTT, 1 μL of RNaseOUT Recombinant RNase Inhibitor, and 1 μL of SuperScript III RT (200 unit/μL) (Invitrogen, Carlsbad, CA, USA) were added. This mixture was incubated at 50 °C for 60 min. The reverse transcriptase was then inactivated by incubation at 70 °C for 15 min. The sample was cooled to 4 °C and stored until further analysis. The primers for the PCR were purchased from Bioneer (Daejeon, Korea) (Table 2).

### 4.7. Western Blot Analysis

After 14 d of cell culture, the cells were lysed with RIPA buffer containing 50 mM Tris-HCl, pH 8.0, 150 mm NaCl, 1% NP-40, 0.5% sodium deoxycholate, 0.1% SDS (Sigma-Aldrich, St Louis, MO, USA), and protease inhibitors (CompleteTM, Roche, Mannheim, Germany). The cell lysates (30 µg of total protein) were then resolved via SDS-polyacrylamide gel electrophoresis and blotted onto nitrocellulose membranes, which were blocked with 5% skim milk in phosphate-buffered saline (PBS) containing 0.2% Tween 20. Immunoblotting analysis was performed according to the manufacturer’s instructions with primary antibodies anti-p-ERK (Cell Signaling, Danvers, MA, USA), anti-bone sialoprotein (Cell Signaling, Danvers, MA, USA), anti-osteoprotegerin (Cell Signaling, Danvers, MA, USA), and anti-osteonectin (Cell Signaling, Danvers, MA, USA). The blots were incubated with the primary antibodies at a dilution of 1:5000, and then further incubated with the appropriate horseradish peroxidase-conjugated secondary antibody.

### 4.8. ALP and Calcium Deposition Assays

ALP deposition was measured using a SensoLyte^®^ para-nitrophenyl phosphate (*p*NPP) Alkaline Phosphatase Assay Kit (ANASPEC, CA, USA). Briefly, collagen samples were prepared from HDPC cultures under various magnitudes of vibration. Following the sample preparation, the samples were mixed with a *p*NPP substrate solution. The mixtures were incubated for 30–60 min, and then a stop solution was added. ALP activity was quantified using absorption at 405 nm.

Calcium deposition was measured using a Quantichrom^TM^ Calcium Assay Kit (Bioassay System, CA, USA). Briefly, the supernatants of the HDPSC cultures were collected. The samples were mixed with the working reagent. The mixtures were then incubated for 3 min, and the amount of calcium in the supernatants was quantified using ELISA at 612 nm.

Mineralized matrix was detected using von Kossa staining. Cultured cells were incubated in 5% silver nitrate (Sigma-Aldrich, St Louis, MO, USA) under ultra-violet light for 60 min, followed by the addition of 3% sodium thiosulphate (Sigma-Aldrich, St Louis, MO, USA) for 5 min and then counterstaining with Van Gieson (Sigma-Aldrich, St Louis, MO, USA) for 5 min. With this staining method, the mineral matrix is stained black and the cell is stained red.

### 4.9. Immunohistochemical Analysis

Cells cultured on cover slides were fixed using 4% paraformaldehyde for 20 min at 4 °C and then washed three times with 10 mM PBS, pH 7.2. These cover slides were then incubated with anti-collagen I (1:500, ab6308, Abcam, Cambridge, UK), anti-osteopontin (1:1000 dilution, ab8448, Abcam, Cambridge, UK), and anti-osteonectin (1:500 dilution, AB 1858, Chemicon, Carlsbad, CA) antibodies for 24 h, followed by signal development using EnVision Plus reagent (Dako, Carpinteria, CA, USA), using diaminobenzidine as the chromogen and Mayer’s hematoxylin as the counterstain.

The relative staining intensity was scored arbitrarily on the light microscopy image as follows: no or weak staining (-), low intensity (1+), moderate intensity (2+), and strong intensity (3+).

### 4.10. Statistical Analysis

Data are expressed as the mean ± SEM of three independent experiments. One-way analysis of variance (ANOVA) followed by the Tukey–Kramer multiple comparisons test were performed with GraphPad Prism (La Kolla, CA, USA). Mean differences were considered significant at *p* < 0.05 (* *p* < 0.05, ** *p* < 0.01, and *** *p* < 0.005). Graphical representations were created using SigmaPlot (Systat Software, Inc., San Jose, CA, USA). All experiments were performed in triplicate.

## 5. Conclusions

In conclusion, our study provides a first glimpse of how mechanical vibration frequency at medium-magnitude (1.12 g) affects the morphology, proliferation, cell cycle, and gene expression pattern of HDPSCs undergoing osteogenic differentiation. The mechanical vibration of 45 Hz is most favorable for HDPSC osteogenic differentiation and decreased cell proliferation, and this effect occurs through the p-ERK/Runx-2 pathway alongside G0/G1 arrest. Studies with animal models are necessary to further investigate this effect in vivo.

## Figures and Tables

**Figure 1 ijms-22-10167-f001:**
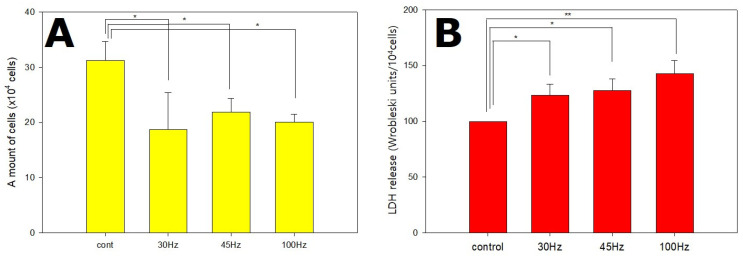
The proliferation (**A**) and lactate dehydrogenase (LDH) secretion (**B**) of human dental pulp stem cells (HDPSCs) subjected to 14 d of mechanical vibrations. HDPSCs were seeded at a density of 2 × 10^4^ cells/well in 6-well plates, and their proliferation was measured after 14 d by counting the viable cells. The growth of the cells was significantly inhibited by mechanical vibration (*n* = 3), and the 100 Hz mechanical vibration group showed higher levels of LDH production than the control (*n* = 6). * *p* < 0.05, ** *p* < 0.01 (compared with the control).

**Figure 2 ijms-22-10167-f002:**
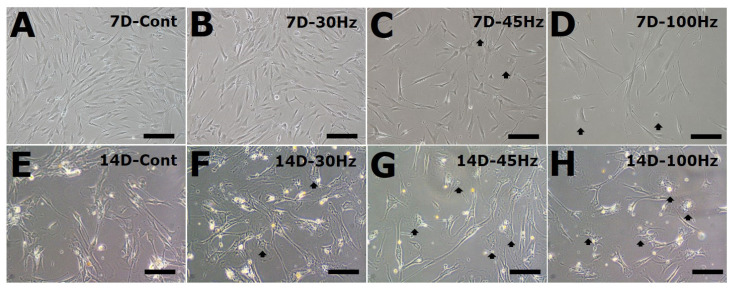
The morphology of human dental pulp stem cells (HDPSCs) subjected to mechanical vibration for 7 D (**A**–**D**) or 14 D (**E**–**G**). Some of the HDPSCs in the vibrated groups assumed a cuboidal shape (**F**–**H**). Black arrows indicating differentiated cells. (**A**) No mechanical vibration (control), (**B**) 30 Hz, (**C**) 45 Hz, (**D**) 100 Hz mechanical vibration. Original magnification; (**A**–**D**) 100×, (**E**–**H**) 200×; Scale bar: (**A**–**D**) 200 μm, (**E**–**H**) 100 μm.

**Figure 3 ijms-22-10167-f003:**
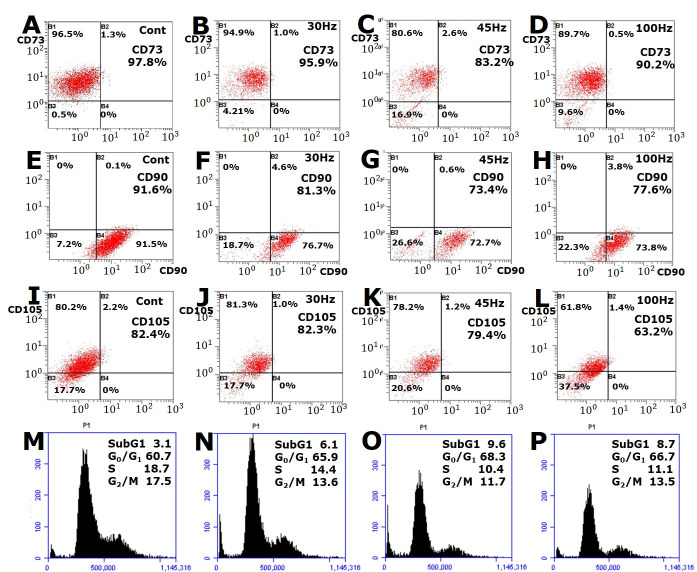
FACS analysis of the surface markers CD 73, CD90, and CD105 after mechanical vibration treatment for 14 d. Human dental pulp stem cells (HDPSCs) were labeled with a FITC- or PE-conjugated antibody and then analyzed in a flow cytometer. CD 73, CD 90, and CD105 expression decreased in the 45- and 100-Hz groups. HDPSCs were subjected to vibration for 7 d, and the cell cycle was assessed using propidium iodide. The percentages of the cells in the G0/G1 phase are indicated.

**Figure 4 ijms-22-10167-f004:**
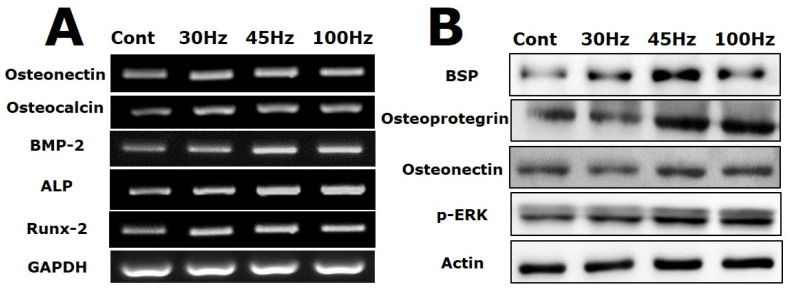
The mRNA (**A**) and protein (**B**) levels of osteogenic markers in human dental pulp stem cells subjected to mechanical vibration for 14 d. Relative mRNA levels of the indicated markers (osteonectin, osteocalcin, BMP-2, ALP, and Runx-2) were measured using Image J software. The mRNA levels of osteocalcin, BMP-2, ALP, and Runx-2 were increased at 45 Hz and 100 Hz, especially osteonectin and Runx-2, whose mRNA levels were significantly higher at 45 Hz than in the control. Quantitation of the p-ERK, BSP, osteoprotegerin, and osteonectin protein levels. The levels of p-ERK, BSP, osteoprotegerin, and osteonectin were increased at 45 Hz compared with the control.

**Figure 5 ijms-22-10167-f005:**
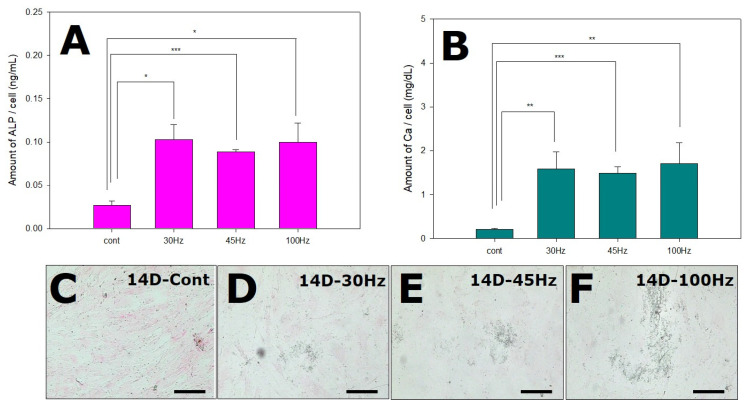
Influence of mechanical vibration on alkaline phosphatase (ALP) production (**A**) and calcium (**B**) and mineral (**C**–**F**) deposition by human dental pulp stem cells. All of mechanical vibration groups showed similarly elevated levels of alkaline phosphatase, and the degree of calcium deposition was increased in the vibrated groups (*n* = 3). Von Kossa staining showing mineral deposition in the cells (**C**–**F**, black spots). * *p* < 0.05, ** *p* < 0.01, *** *p* < 0.005, (**C**–**F**) von Kossa staining, Original magnification: 200×, Scale bar: 100 μm.

**Figure 6 ijms-22-10167-f006:**
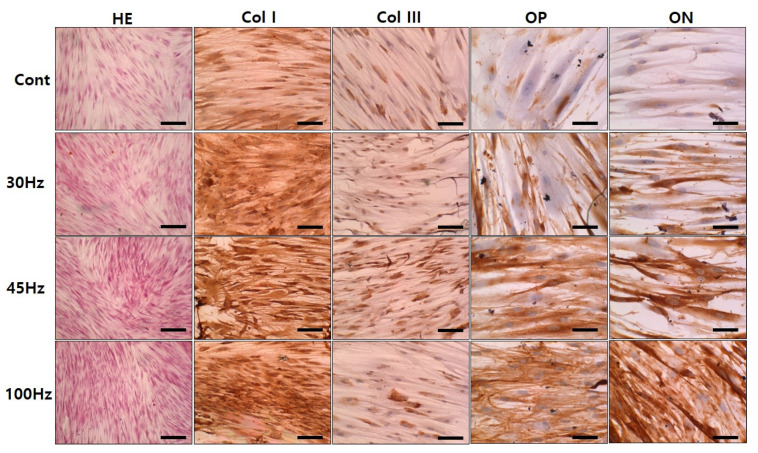
Immunohistochemical staining of human dental pulp stem cells subjected to 14 d of mechanical vibrations. Collagen type I was strongly detected at 45 Hz and 100 Hz, and collagen type III was strongly detected at 45 Hz. Additionally, osteopontin (OP) and osteonectin (ON) were strongly detected at 45 Hz and 100 Hz. HE: Hematoxylin-eosin staining. Original magnification; HE: 100×, Col I and Col III: 200×, OP and ON: 400×, Scale bar; HE: 200 μm, Col I and Col III: 100 μm, OP and ON: 50 μm.

**Figure 7 ijms-22-10167-f007:**
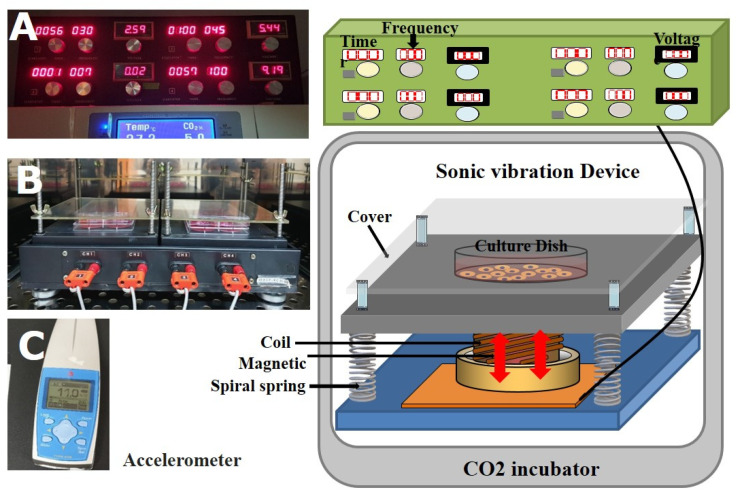
The schematic diagram of the mechanical vibration culture and photographs of the controller (**A**) and vibrator (**B**). The controller regulates the intensity, frequency, and operation times (**C**). The four channels of the vibrator were made using the principles of speakers.

**Table 1 ijms-22-10167-t001:** Relative staining intensity score for coll I, coll III, osteopontin and osteonectin.

Marker	Coll I	Coll III	Osteopontin	Osteonectin
Cont	+	+	+	-
30 Hz	++	+	++	+
45 Hz	++	++	+++	++
100 Hz	++	+	+++	+++

**Table 2 ijms-22-10167-t002:** Reverse transcriptase-polymerase chain reaction (RT-PCR) primers sequences.

Gene	Forward (5′-3′)	Reverse (5′-3′)	Product Size	NCBI Accession Number
GAPDH	ACC ACA GTC CAT GCC ATC AC	TCC ACC ACC CTG TTG CTG TA	452	NM_001357943.2
Osteonectin	CCA GAA CCA CCA CTG CAA AC	GGC AGG AAG AGT CGA AGG TC	155	NM_001309444.2
Osteocalcin	CCA GGC GCT ACC TGT ATC AA	AGG GGA AGA GGA AAG AAG GG	231	NM_199173.6
BMP2	GTA CTA GCG ACA CCC ACA AC	GTC CAG CTG TAA GAG ACA CC	316	NM_001200.4
ALP	ATCTCGTTGTCTGAGTACCAGTCC	TGGAGCTTCAGAAGCTCAACACCA	454	NM_001127501.4
Runx2	ACA GTA GAT GGA CCT CGG GA	ATA CTG GGA TGA GGA ATG CG	113	NM_001015051.4

BMP2; bone morphogenetic protein 2, ALP; alkaline phosphatase, Runx2; runt-related transcription factor 2.

## Data Availability

The data generated and analyzed during this study are available from the corresponding author on reasonable request.

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
