# Peer review of "The Osteogenic Differentiation of Human Dental Pulp Stem Cells through G0/G1 Arrest and the p-ERK/Runx-2 Pathway by Sonic Vibration"

_ijms, 2021, doi:10.3390/ijms221810167_

Round 1

Reviewer 1 Report

This manuscript describes the effects of medium-magnitude (1.12 g) sonic vibration with a frequency of 30, 45, or 100 Hz on human dental pulp stem cells (HPSCs) during 14 days of osteogenic differentiation. Vibration with each frequency was found to decrease proliferation of HPSCs. Vibration with either 45 Hz or 100 Hz reduced expression of mesenchymal-associated surface markers in HPSCs. Gene and protein expression levels of osteogenic markers were higher in HPSCs subjected to vibration with 45 Hz and 100 Hz compared to static control and HPSCs subjected to vibration with 30 Hz. Moreover, vibration with each frequency was found to increase ALP activity of HPSCs. The manuscript is well-written. However, there are several comments that the authors need to address.

1. The authors should provide a range of magnitude for low-, medium-, and high-magnitude vibration in the introduction.

2. What are the advantages of using medium-magnitude vibration in enhancing osteogenesis compared to low- and high-magnitude vibration?

3. Why 45 Hz was better than 30 Hz and 100 Hz in enhancing osteogenesis? The authors should include their replies to this comment in the discussion.

4. Figure 1B: I believe that there were significant differences in terms of LDH release between control and 45 Hz as well as 100 Hz. Please revise the figure.

5. Line 457: The details of primers for each studied gene (PCR product length, primer sequence, and NCBI accession number) should be provided.

6. The authors should add a statistical analysis section in the materials and method.

7. The following relevant works should be cited and discussed.

- Recent advances in mechanically loaded human mesenchymal stem cells for bone tissue engineering (2020) International Journal of Molecular Sciences 21(16): 5816.

- Acoustic-frequency vibratory stimulation regulates the balance between osteogenesis and adipogenesis of human bone marrow-derived mesenchymal stem cells (2015) BioMed Research International 2015: 540731.

Author Response

  • The authors should provide a range of magnitude for low-, medium-, and high-magnitude vibration in the introduction.

Ans: Thank you very much for your good comment. We provide a range of magnitude as your comments

Many studies have addressed the effects of vibration stimulation on the differentiation of various types of stem cells. Some researchers defined low-magnitude as <1 gravity (g = acceleration of 9.81 m/s2) and 3 ~ 5 g as high-magnitude [36], but other researchers described a range of 0.3 to 6 g as low-magnitude [37]. Thus, the definitions of low- and high-magnitude have not been standardized. There are also few reports about the effects of 1.12 g magnitude sonic vibrations on the osteogenic differentiation of human dental pulp stem cells (HDPSCs). In our study, we defined 1 to 2.9 g as medium-magnitude.

  • What are the advantages of using medium-magnitude vibration in enhancing osteogenesis compared to low- and high-magnitude vibration?

Ans: Thank you very much for your good questions.

In a related study, researchers induced osteogenesis with 0.15, 1, and 2g of magnitude. They had showed that 2g- magnitude reduced in cell proliferation, Runx-2, and collagen expression [33]. Also, in our unpublished prior work, cell activity in magnitude of more than 1.5 g was reduced and stress increased. Therefore, in our study was conducted with magnitude of less than 1.5 g.

  • Why 45 Hz was better than 30 Hz and 100 Hz in enhancing osteogenesis? The authors should include their replies to this comment in the discussion.

Ans: Thank you very much for your good comment. We added an explanation to your comment.

In our study, based on the western blot, fluorescence, and CD marker results, we can conclude that differentiation was more strongly induced at 45 and 100 Hz. The proportion of cells in G0/G1 was the highest at 45 Hz. Also, the control group was observed to be in the proliferative phase and the proportion of cells in G2/M was the highest in the control 17.5% (Fig. 3). This suggests that sonic vibration affects the cell cycle and reduces proliferation by arresting the cell cycle, an effect more pronounced at 45 Hz compared to the other groups.

  1. Figure 1B: I believe that there were significant differences in terms of LDH release between control and 45 Hz as well as 100 Hz. Please revise the figure.

Ans: Thank you very much for your good point out. I revised figure 1 like your comments.

Figure 1. The proliferation (A) and lactate dehydrogenase (LDH) secretion (B) of human dental pulp stem cells (HDPSCs) subjected to 14 d of mechanical vibrationd. HDPSCs were seeded at a density of 2 × 104 cells/well in 6-well plates, and their proliferation was measured after 14 d by enumeration of viable cells. The growth of cells was significantly inhibited by mechanical vibration (n = 3), and the 100-Hz mechanical vibration group showed higher levels of LDH production than the control (n = 6). *P < 0.05, **P < 0.01, ***P < 0.005 (compared with the control)

  1. Line 457: The details of primers for each studied gene (PCR product length, primer sequence, and NCBI accession number) should be provided.

Ans: Thank you for your good point out. I provided table like your point out.

Table 1 Reverse transcriptase-polymerase chain reaction (RT-PCR) primers sequence

Gene

Forward (5’-3’)

Reverse(5’-3’)

Product

size

NCBI accession

number

GAPDH

ACC ACA GTC CAT GCC ATC AC

TCC ACC ACC CTG TTG CTG TA

452

NM_001357943.2 

Osteonectin

CCA GAA CCA CCA CTG CAA AC

GGC AGG AAG AGT CGA AGG TC

155

NM_001309444.2

Osteocalcin

CCA GGC GCT ACC TGT ATC AA

AGG GGA AGA GGA AAG AAG GG

231

NM_199173.6

BMP2

GTA CTA GCG ACA CCC ACA AC

GTC CAG CTG TAA GAG ACA CC

316

NM_001200.4

ALP

ATCTCGTTGTCTGAGTACCAGTCC

TGGAGCTTCAGAAGCTCAACACCA

454

NM_001127501.4

Runx2

ACA GTA GAT GGA CCT CGG GA

ATA CTG GGA TGA GGA ATG CG

113

NM_001015051.4

BMP2; bone morphogenetic protein 2, ALP; alkaline phosphatase, Runx2; runt-related transcription factor 2

  1. The authors should add a statistical analysis section in the materials and method.

Ans: Thank you for your point out. We added a statistical analysis as your point out.

4.10. Statistical Analysis

Data are expressed as the mean ± SEM of three independent experiments. One-way analysis of variance (ANOVA) followed by the Tukey-Kramer multiple comparisons test was performed with GraphPad Prism (La Kolla, California, USA). Mean differences were considered significant at P < 0.05 (*P < 0.05, **P < 0.01, and ***P < 0.005). Graphical representations were created using SigmaPlot (Systat Software, Inc., San Jose, CA, USA). All experiments were performed in triplicate.

  1. The following relevant works should be cited and discussed.

Ans: Thank you very much for your good advices. We did the citation and added an discuss to the manuscript as your advices.

- Recent advances in mechanically loaded human mesenchymal stem cells for bone tissue engineering (2020) International Journal of Molecular Sciences 21(16): 5816.

- Acoustic-frequency vibratory stimulation regulates the balance between osteogenesis and adipogenesis of human bone marrow-derived mesenchymal stem cells (2015) BioMed Research International 2015: 540731.

Especially, mechanical/physical stimulation, such as perfusion, stretching, vibration, and compression have been shown to affect the functions of many cell types [36-38]. Also, many researchers have studied the effects of significant variables such as acceleration, flow rate, frequency, periods, strain, and load [38].

Chen et al. showed that 800 Hz sonic vibration (0.30 g) increased the osteogenic differentiation of human MSCs more than 30 and 400 Hz, as assessed via alizarin red staining and mRNA expression analysis [39].

Reviewer 2 Report

Comments:

The study “The osteogenic differentiation of human dental pulp stem cells through G0/G1 arrest and the p-ERK/Runx-2 pathway by sonic vibration” has provided that they showed that vibration of 45-Hz frequency at a medium-magnitude induces the G0/G1 arrest of HDPSCs through the p-ERK/ Runx-2 pathway and can serve as a potent stimulator for their differentiation and extracellular matrix production. The following questions are concerned:

  1. This manuscript has many problems with writing, and grammar, it needs to be carefully checked and corrected. The authors should consult a native English speaker before submitting the manuscript.
  2. How do authors define the low, medium, and strong sonic vibration in the treatments?
  3. Line 99-100, please the cell number present as the mean ± standard deviation (SD)
  4. Why are no significant of 45Hz and 100Hz groups as compared with control in the Figure 1B?
  5. Line 109 and line 111, please indicates as day 7 and day 14.
  6. Line 116 and line 117, please indicates as day 7 and day 14. What do the arraws indicate in Figure 2?
  7. Line 129, Figure 3. The sentence is confused as “mechanical vibration Table 14. d.”. No negative cell surface markers in figure 3.
  8. Table 1 is The data already show in figure 3.
  9. In the section 2.4., what is the significant of inhibition of proliferation and G0/G1 arrest in HDPSCs by sonic vibration? Dose G0/G1 arrest cause cell apoptosis?
  10. The description of section 2.5 is hard to read and confuse. Please verified. Part of descriptions should move to Discussion section (line 160-164, line 175-177, line 181-184, and line 191-194).
  11. In the figure 4, GAPDH and actin should move to last one.
  12. What are the experimental groups in C, D, E, F of figure 5?
  13. The higher level or strong expression are ambiguous. Please quantify the results from the figure 6.
  14. Why do authors chose the medium-magnitude sonic vibration for the study?
  15. There are to many sections in the Discussion. Please verified and the contexts should describe clearly.

Author Response

  • This manuscript has many problems with writing, and grammar, it needs to be carefully checked and corrected. The authors should consult a native English speaker before submitting the manuscript.

Ans: According to your comments, I asked a native English speaker to correct this manuscript for grammar and sentences.

  1. How do authors define the low, medium, and strong sonic vibration in the treatments?

Ans: Thank you very much for your good comment. We provide a range of magnitude as your comments

Many studies have addressed the effects of vibration stimulation on the differentiation of various types of stem cells. Some researchers defined low-magnitude as <1 gravity (g = acceleration of 9.81 m/s2) and 3 ~ 5 g as high-magnitude [36], but other researchers described a range of 0.3 to 6 g as low-magnitude [37]. Thus, the definitions of low- and high-magnitude have not been standardized. There are also few reports about the effects of 1.12 g magnitude sonic vibrations on the osteogenic differentiation of human dental pulp stem cells (HDPSCs). In our study, we defined 1 to 2.9 g as medium-magnitude.

  1. Line 99-100, please the cell number present as the mean ± standard deviation (SD).

Ans: Thank you for your good point out. We provided a standard deviation for cell number as your point out.

The growth of the cells was significantly inhibited in the mechanical vibration group (Figure 1A). The cell number was determined to be approximately 3.1±0.32 × 105 in the control group, 1.9±0.79 × 105 in the 30 Hz group, 2.1±0.26 × 105 in the 45 Hz group, and 2.0±0.11 × 105 in the 100 Hz group after 14 d of culture.

  • Why are no significant of 45Hz and 100Hz groups as compared with control in the Figure 1B?

Ans: Thank you for your good comment. We analyzed the statistics of the graph results. And we revised the graph as follow.

Figure 1. The proliferation (A) and lactate dehydrogenase (LDH) secretion (B) of human dental pulp stem cells (HDPSCs) subjected to 14 d of mechanical vibrationd. HDPSCs were seeded at a density of 2 × 104 cells/well in 6-well plates, and their proliferation was measured after 14 d by enumeration of viable cells. The growth of cells was significantly inhibited by mechanical vibration (n = 3), and the 100-Hz mechanical vibration group showed higher levels of LDH production than the control (n = 6). *P < 0.05, **P < 0.01, ***P < 0.005 (compared with the control)

  1. Line 109 and line 111, please indicates as day 7 and day 14.

Ans: Thank you for your good point out. We inserted the date as you pointed out.

Figure 2 shows the morphology of the HDPSCs during osteogenesis at different frequencies on d 7 and d 14. Most of the cells were short spindle and narrow fibroblast-like (Fig. 2A-D), but some of the cells were cuboidal (black arrow) (Fig. 2C and D) on day 7. On d 14 of culture, many of the cell morphology had changed to a cuboidal shape (black arrow) at 45 Hz and 100 Hz compared to the control groups. These morphological changes indicate that vibration of 45 Hz and 100 Hz induced more osteogenic differentiation compared with the 30 Hz and control groups (Fig. 2).

  • Line 116 and line 117, please indicates as day 7 and day 14. What do the arraws indicate in Figure 2?

Ans: Thank you for your good point out. We inserted the date as you pointed out.

Figure 2. The morphology of human dental pulp stem cells (HDPSCs) subjected to mechanical vibration for 7 D (A-D) or 14 D (E-G). Some of the HDPSCs in the vibrated groups assumed a cuboidal shape (F-H). Black arrows indicating differentiated cells. A: No mechanical vibration (control), B: 30 Hz, C: 45 Hz, D: 100 Hz mechanical vibration. Original magnification; A-D: 100×, E-H: 200×; Scale bar: A-D: 200 μm, E-H: 100 μm

Usually, HDPSC morphology is fibroblast-like, but some of the cells changed to a cuboidal or polygonal shape in the mechanical vibration groups at 14 d. It was previously reported that DPSCs showed a fibroblastic-elongated morphology at 7 d in differentiation media [27], and other studies have reported altered polygonal morphology of DPSCs after 21 d in osteogenic differentiation media [28]. In addition, Demiray et al. have shown that a polygonal morphological change can be induced by mechanical vibrations (90 Hz, 0.15 g) of mouse MSCs during osteogenesis [10]. However, in this study, such a morphological change was not as dramatic, perhaps due to the short cell differentiation time.

  • Line 129, Figure 3. The sentence is confused as “mechanical vibration Table 14. d.”. No negative cell surface markers in figure 3.
  1.  

In this study, we did not proceed with the analysis of negative cells that did not induce differentiation. . In our previous study, isolated cells expressed MSC surface antigens including CD73 (99.88%), CD90 (99.81%), and CD105 (99.87%) (Data not shown). The disappearance of CD73, CD90, and CD105 antigen expression during osteogenesis suggests that these proteins may be involved in the regulation of osteogenesis.

In Figure, CD73 and CD105 are showed on the Y axis and CD90 on the X axis. So we added new labelling to the graph.

  • Table 1 is The data already show in figure 3.

Ans: Thank you for your advice. We deleted the table 1 as your advice.

  • In the section 2.4., what is the significant of inhibition of proliferation and G0/G1 arrest in HDPSCs by sonic vibration? Dose G0/G1 arrest cause cell apoptosis?

Ans: Thank you for your good question. We explained about your question as follow.

In our study, based on the western blot, fluorescence, and CD marker results, we can conclude that differentiation was more strongly induced at 45 and 100 Hz. The proportion of cells in G0/G1 was the highest at 45 Hz. Also, the control group was observed to be in the proliferative phase and the proportion of cells in G2/M was the highest in the control 17.5% (Fig. 3). This suggests that sonic vibration affects the cell cycle and reduces proliferation by arresting the cell cycle, an effect more pronounced at 45 Hz compared to the other groups. In the PI staining results, sub-G1, which is generally thought to indicate apoptosis, was increased at 45 Hz and 100 Hz. However, some researchers have reported that sub-G1 may also indicate the cells are differentiating [40]. We did not observe apoptosis or necrosis of the cells under a microscope.

Given the established facts that cell-cycle progression and differentiation are two distinct and mutually exclusive processes [31] and that increases in cell differentiation are generally accompanied by a parallel reduction in cell growth [32], the arrest of cell growth by vibration implies that it can accelerate the osteogenic differentiation of HDPSCs.

  • The description of section 2.5 is hard to read and confuse. Please verified. Part of descriptions should move to Discussion section (line 160-164, line 175-177, line 181-184, and line 191-194).

Ans: Thank you for your advice. We verified and move to discussion as your advice.

  • In the figure 4, GAPDH and actin should move to last one.

Ans: Thank you for your advice. GAPDH has been moved down as your advice.

  • What are the experimental groups in C, D, E, F of figure 5?

Ans: Thank you for your good point out. We revised for the C-F of figure.

Figure 5. Influence of mechanical vibration on alkaline phosphatase (ALP) production (A) and calcium (B) and mineral (C-F) deposition by human dental pulp stem cells. All of mechanical vibration groups showed similarly elevated levels of alkaline phosphatase, and the degree of calcium deposition was increased in the vibrated groups (n = 3). Von Kossa staining showing mineral deposition in the cells (C-F, black spots). *p < 0.05, C-F: von Kossa staining, Original magnification: 200×, Scale bar: 100 μm

  • The higher level or strong expression are ambiguous. Please quantify the results from the figure 6.

Ans: Thank you very much for your good comment. We have summarized the degree of expression as follows.

The relative staining intensity was scored arbitrarily on the light microscopy image as follows: no or weak staining (-); low intensity (1+), moderate intensity (2+), and strong intensity (3+).

Table 1. Relative staining intensity score for coll I , coll III, osteopontin and osteonectin

Marker

coll I

coll III

osteopontin

osteonectin

Cont

+

+

+

-

30 Hz

++

+

++

+

45 Hz

++

++

+++

++

100 Hz

++

+

+++

+++

  • Why do authors chose the medium-magnitude sonic vibration for the study?

Ans: Thank you very much for your good questions.

In a related study, researchers induced osteogenesis with 0.15, 1, and 2g of magnitude. They had showed that 2g- magnitude reduced in cell proliferation, Runx-2, and collagen expression [33]. Also, in our unpublished prior work, cell activity in magnitude of more than 1.5 g was reduced and stress increased. Therefore, in our study was conducted with magnitude of less than 1.5 g.

  • There are to many sections in the Discussion. Please verified and the contexts should describe clearly.

Ans: Thank you very much for your good advice. We revised the discussion as your comments.

Reviewer 3 Report

Recommendations: Major Revision

The authors have studied the effect of medium-magnitude (1.12 16 g) sonic vibration on the osteogenic differentiation of human dental pulp stem cells frequency 18 of 30, 45, or 100 Hz. The cells were in vitro studies such as cell proliferation, LDH assay, FACS, cell cycle analysis,  RT-PCR, western blot analysis, and immunohistochemical analysis. The authors concluded that mechanical vibration frequency at medium-magnitude showed proliferation, cell cycle, and gene expression, and mechanical vibration at 45 Hz was most favorable for osteogenic differentiation of HDPSCs cells. In the current reviewer’s opinion, the manuscript may be considered for a major revision.

Major comments:

 Comment 1: I recommend the author add more details in the introduction part to explain why there is a need for HDPSCs by ultra vibration compared to other simulation methods. Also, authors may include more information on HDPSCs cells using sonic vibrations.

Comment 2: In the materials and method section, authors may give full details of proliferation experiments and LDH assay experiments details.  In their current state, it is difficult to understand the experiments. It is recommended to provide the full details of the experiments for a better understanding of the readers.

Comment 3: I recommend the author improve the formatting and quality of figure 3 as the figures are too small and increasing font size of x and y-axis.

 Comment 4: The authors may increase the timepoint up to days 21 and 28 for calcium deposition compared to days 7 and 14. Also, authors may used positive control BMP-2 to compare sonic vibration for cells to differentiate into osteoblast.

Comment 5: I recommend that the authors provide in vitro characterization to support the evidence for osteogenic properties. Moreover, in vitro cell studies provide more evidence in terms of the cytotoxicity of the cells. This can be resolved by performing perform osteogenesis characterizations such as DNA assay, alizarin red staining.

Author Response

Major comments:

Comment 1: I recommend the author add more details in the introduction part to explain why there is a need for HDPSCs by ultra vibration compared to other simulation methods. Also, authors may include more information on HDPSCs cells using sonic vibrations.

Ans: Thank you very much for your good comment. We revised the introduction and explain about as your comments.

Healthcare products using a variety of mechanical/physical stimuli have been commercialized, such as sonic vibrating tooth brushes, and studies of their effects are being conducted. One investigator reported that various frequencies and intensities of sonic vibrations stimulate the neurological system, bone, spine, and intervertebral discs [36]. Chen et al. showed that 800 Hz sonic vibration (0.30 g) increased the osteogenic differentiation of human MSCs more than 30 and 400 Hz, as assessed via alizarin red staining and mRNA expression analysis [39]. Also, osteocytes subjected to sonic vibration (0.3 g) upregulate COX-2 expression 3-fold at 90 Hz but downregulate RANKLE expression by half at 60 Hz [13]. Tirkkonen et al. have reported that 100 Hz sonic vibration (3.0 g) induces the osteogenic differentiation of human adipocyte stem cells by more than 50 Hz, as assessed via alizarin red staining and collagen expression analysis [14]. These results suggest that osteogenesis and bone resorption can be modulated through specific vibration-induced changes in cell activity, which depend on the vibration frequency and occur via differential gene expression. However, there have been few reports of HDPSCs subjected to sonic vibrations.

Comment 2: In the materials and method section, authors may give full details of proliferation experiments and LDH assay experiments details. In their current state, it is difficult to understand the experiments. It is recommended to provide the full details of the experiments for a better understanding of the readers.

Ans: Thank you for your good advice. We revised and re-write for method as your advice.

Cell proliferation was measured using an automated cell counter (Scepter™, Millipore Corporation, Billerica, MA, USA). Attach a sensor probe to the end of the Scepter unit with the electrode sensing panel facing toward the front of the device.

The cultured cells were washed with PBS and detached with accutase for 10 min. After centrifugation at 1000 rpm, the supernatant was discarded and the cell pellet was resuspended in 1 ml PBS. And about 50 µl of the sample is taken up by the probe of the Scepter. Then the device will display a histogram of the cell size or diameter on its screen as well as the cell concentration per ml.

LDH activity was measured using an Lactate Assay Kit (Abcam, Cambridge, UK) according to the instructions of the manufacturer. In 1st step, the frozen reagent was thawed. Thaw the lactate probe solution in a 37 °C water bath for 1 - 5 min and lactate enzyme mix dissolve in 220 µL lactate assay buffer. And 2nd step, we prepared a reaction solution. This reaction solution is composed of 46 µl of lactate assay buffer, 2 µl of probe, and 2 µl of enzyme. After 14 d of culture, 50 µl of media and 50 µl of the reaction solution were mixed and incubated in the dark at room temperature for 30 min. The reaction was terminated by adding the stop solution (1 N HCl), and the absorbance was measured at 570 nm.

Comment 3: I recommend the author improve the formatting and quality of figure 3 as the figures are too small and increasing font size of x and y-axis.

Ans: Thank you for your good point out. I revised figure 3 like your comments.

Figure 3. FACS analysis of the surface markers, CD 73, CD90 and CD105 after mechanical vibration treatment for 14 days. HDPSCs were labeled with FITC- or PE-conjugated antibody and then analyzed in a flow cytometer. The CD73, CD90 and CD105 expression decreased in the 45 and 100 Hz groups. Human dental pulp stem cells were treated with vibration for 7 days and the cell cycle phase was detected using a PI. The percentages of G0/G1 cells are indicated.

Comment 4: The authors may increase the timepoint up to days 21 and 28 for calcium deposition compared to days 7 and 14. Also, authors may used positive control BMP-2 to compare sonic vibration for cells to differentiate into osteoblast.

Ans: Thank you for your good comment. As your comment, I am going to proceed with the results of the 28 day experiment. We can't do the experiment because it's the 10 days revision period. Please give us a four-week experimental period for the second revision.

Comment 5: I recommend that the authors provide in vitro characterization to support the evidence for osteogenic properties. Moreover, in vitro cell studies provide more evidence in terms of the cytotoxicity of the cells. This can be resolved by performing perform osteogenesis characterizations such as DNA assay, alizarin red staining.

Ans: Thank you for your good comment. As your comment, I am going to analysis for with the DNA assay and alizarin red staining. Please give us a four-week experimental period for the second revision.

Round 2

Reviewer 1 Report

The quality of the manuscript has been greatly improved. I have no further comment.

Author Response

Thank you very much for your good comment and advice. We were able to revise the contents that we didn't think of with your help. I'll take a closer look at this in the future. Thanks once again.

Reviewer 2 Report

Please verified the figure 4B. The actin is a internal control, please put it at the last one as the GAPDH in figure 4A. Osteonectin, the first word need capital.

Author Response

Ans: Thank you very much for your point out. We revised the figure 4B as your pointed out.

We were able to revise the contents that we didn't think of with your help. I'll take a closer look at this in the future. Thanks once again.

Reviewer 3 Report

The manuscript may be accepted after the comparing the results with positive control. Thanks

Author Response

The manuscript may be accepted after the comparing the results with positive control. Thanks

Comment 4: The authors may increase the timepoint up to days 21 and 28 for calcium deposition compared to days 7 and 14. Also, authors may used positive control BMP-2 to compare sonic vibration for cells to differentiate into osteoblast.

Ans: Thank you for your good comment. As your comment, I am going to proceed with the results of the 28 day experiment. We can't do the experiment because it's the 10 days revision period. Please give us a four-week experimental period for the second revision.

Comment 5: I recommend that the authors provide in vitro characterization to support the evidence for osteogenic properties. Moreover, in vitro cell studies provide more evidence in terms of the cytotoxicity of the cells. This can be resolved by performing perform osteogenesis characterizations such as DNA assay, alizarin red staining.

Ans: Thank you for your good comment. As your comment, I am going to analysis for with the DNA assay and alizarin red staining. Please give us a four-week experimental period for the second revision.
